# Comparison of alemtuzumab, anti-thymocyte globulin, and post-transplant cyclophosphamide for graft-versus-host disease and graft-versus-leukemia in murine models

Kiyomi Mashima[1], Iekuni Oh[1], Ken Fujiwara[2], Junko Izawa[1], Norihito Takayama[1], Hirofumi Nakano[1], Yasufumi Kawasaki[1], Daisuke Minakata[1], Ryoko Yamasaki[1], Kaoru Morita[1], Masahiro Ashizawa[1], Chihiro Yamamoto[1], Kaoru Hatano[1], Kazuya Sato[1], Ken Ohmine[1], Shin-Ichiro Fujiwara[1], Nobuhiko Ohno[2], Yoshinobu Kanda[1]*

1 Division of Hematology, Department of Medicine, Jichi Medical University, Tochigi, Japan, 2 Division of Histology and Cell Biology, Department of Anatomy, School of Medicine, Jichi Medical University, Tochigi, Japan

* ycanda-tky@umin.ac.jp

**Data Availability Statement:** All relevant data are within the manuscript and its Supporting Information files.

## Abstract

Graft-versus-host disease is a major complication after allogeneic hematopoietic stem cell transplantation for hematological malignancies. Immunosuppressive drugs, such as anti-thymocyte globulin, alemtuzumab, and post-transplant cyclophosphamide, have been used to prevent graft-versus-host disease in HLA-mismatched haploidentical hematopoietic stem cell transplantation. Here, we investigated whether these drugs could ameliorate graft-versus-host disease without diminishing the graft-versus-leukemia effect by using a xenogeneic transplanted graft-versus-host disease/graft-versus-leukemia model. Anti-thymocyte globulin treatment diminished graft-versus-host disease symptoms, completely depleted the infiltration of inflammatory cells in the liver and intestine, and led to prolonged survival. By contrast, improvement after post-transplant cyclophosphamide treatment remained minimal. Alemtuzumab treatment modestly prolonged survival despite an apparent decrease of Tregs. In the graft-versus-leukemia model, 1.5 to 2.0 mg/kg of anti-thymocyte globulin and 0.6 to 0.9 mg/kg of alemtuzumab reduced graft-versus-host disease with minimal loss of graft-versus-leukemia effect. Mice treated with 400 mg/kg of post-transplant cyclophosphamide did not develop graft-versus-host disease or leukemia, but it was difficult to evaluate the graft-versus-leukemia effect due to the sensitivity of A20 cells to cyclophosphamide. Although the current settings provide narrow optimal therapeutic windows, further studies are warranted to maximize the benefits of each immunosuppressant.

## Introduction

An important effect of allogeneic hematopoietic stem cell transplantation (HSCT) for hematological malignancies is the profound alloimmune response, which is called the graft-versus-

**Funding:** Kiyomi Mashima, Jichi Medical University Graduate Student Start-up Award, name of the agency: Jichi Medical University, grant number: we have no grant number, a description of each funder's role: the funders had no role in study design, data collection and analysis, decision to publish, or preparation of the manuscript.

**Competing interests:** The authors have declared that no competing interests exist.

leukemia (GVL) effect; this effect is mediated by donor T cells. However, alloimmune responses also provoke graft-versus-host disease (GVHD), which is a serious complication following HSCT. The combination of calcineurin inhibitors with methotrexate has been widely used for prophylaxis against GVHD. However, in HSCT from HLA-mismatched haploidentical donors, more potent immunosuppressive drugs, such as anti-thymocyte globulin (ATG), alemtuzumab (recombinant human anti-CD52 monoclonal antibody), and post-transplant cyclophosphamide (PTCY), have been investigated. No clinical trials have directly compared these three drugs.

Some retrospective analysis have compared the immunosuppressive effects of alemtuzumab, ATG, and PTCY for GVHD prophylaxis. Patients who underwent HSCT from an unrelated donor and received 5 mg/kg of ATG showed higher rates of both acute and chronic GVHD than haploidentical HSCT recipients who received 100 mg/kg of PTCY [1]. Moreover, among haploidentical transplant recipients, 10 mg/kg of ATG was associated with a higher rate of grade III-IV acute GVHD compared to 50 mg/kg of PTCY for two days [2]. Another retrospective analysis showed that alemtuzumab at a dose of 60 to 100 mg/body decreased the risk of both acute and chronic GVHD more strongly than 5 or 10 mg/kg of ATG, although at the expense of increased incidences of relapse and virus infection. Alemtuzumab did not increase the risk of PTLD compared to ATG since it suppressed not only T cells but also B cells [3]. The appropriate strategy for GVHD prophylaxis after mismatched HSCT remains controversial.

To evaluate the role of human specific anti-human cell blocking antibodies *in vivo*, we used humanized mice (human MNC → NOG mouse). We investigated whether these drugs could ameliorate GVHD without diminishing the GVL effect by using a xenograft GVHD model, in which we infused human peripheral blood mononuclear cells (hPBMCs) into NOG mice to evaluate the effects of immunosuppressants against human cells.

## Materials and methods

### Cells

hPBMCs from healthy adult volunteers were isolated by density gradient centrifugation using Lymphoprep™ (Axis Shield, UK) after obtaining written informed consent from each volunteer as in our previous report [4]. This study was approved by the Institutional Review Board of Jichi Medical University and was conducted according to the principles of the Declaration of Helsinki. Firefly luciferase-transfected A20 BALB/c strain mouse B leukemia and lymphoma cells were kindly gifted by Dr. K. Ohnuma in October 2017. (Juntendo University, Tokyo, Japan) [5].

### Mice

Female NOD/Shi-scid/IL2R $\gamma^{null}$ mice (NOG) were purchased from the Central Institute of Experimental Animals (Kawasaki, Japan) and housed in our mouse facility at Jichi Medical University. All mice used in the experiments were 8–12 weeks old. We performed our animal experiments as described before [6]. In brief, the animals were maintained under a 12-hr light/dark cycle and given conventional food and water ad libitum in 23°C room. The animals were anesthetized with pentobarbital sodium (30 mg/kg intraperitoneally, i.p., Kyoritsu Seiyaku, Tokyo, Japan). They were euthanized when body weight loss reached 15% within a few days or an overall body weight loss reached 20%. All animal protocols were approved by the Animal Care and Use Ethics Committee in our institute.

## Xenogeneic GVHD models and GVHD prophylaxis using immunosuppressants

The NOG mice were irradiated with 2 Gy (gamma irradiator with a Cesium[137] source) and injected i.v. with $5 \times 10^6$ hPBMCs suspended in 500 µL of Dulbecco's phosphate-buffered saline (DPBS) on the day of transplantation. Each time, we used three mice per group and repeated the experiment two to three times. The experiments in all the groups were performed at the same time with a single healthy donor. We used different donors in the experiments at different time.

For the GVHD prophylaxis, ATG (Sanofi, Tokyo, Japan) (rabbit-derived antibodies against human thymocytes), alemtuzumab (Sanofi, Tokyo, Japan) (humanized monoclonal antibodies against human CD52), or PTCY were used. ATG powder was dissolved in DPBS and adjusted to a concentration of 1 mg/mL. Alemtuzumab (30 mg/mL) was diluted 2,000-fold with PBS just before being administered to the mice. Cyclophosphamide was dissolved in pure water and adjusted to 10 mg/mL with DPBS just before use. After the immunosuppressant concentrations were adjusted, the dosages of these solutions were adjusted relative to the mouse body weight. PBS was used to adjust the final volume to 500 µL for each mouse. The ATG and alemtuzumab solutions were intraperitoneally injected into the mice on days -4 and -3, and the cyclophosphamide solution was administered in the same manner on days 3 and 4. All mice were examined daily for survival, and body weight was examined every other day. Observation was discontinued at 28 days in the GVHD experiments.

## GVL models and bioluminescence imaging

We used the A20 mouse leukemic cell line (BALB/c, strain B cell leukemia/lymphoma) transduced with luciferase as a model for leukemia. For the GVL model, A20 cells ($2 \times 10^3$) and hPBMCs ($5 \times 10^6$) were co-infused following irradiation. Prophylactic treatment for GVHD was performed in the same manner as in the GVHD experiments using the three immunosuppressants. All mice were examined daily for survival, and clinical GVHD scores were examined every other day. The clinical GVHD scores were calculated based on body weight, activity, skin, fur ruffing, and posture. Each factor received 0 to 2 scores, and the total score was determined by sum them (maximum index was 10.) as we previously reported [4]. The observation period lasted 50 days.

D-luciferin sodium salt (OZ Biosciences, Marseille, France) was dissolved in DPBS to a final concentration of 30 mg/mL in accordance with the instructions and stored at -80°C. Each mouse was injected with 100 µL of D-luciferin stock substrate solution 10 minutes before *in vivo* imaging with an IVIS Spectrum CT In Vivo Imaging System (Perkin Elmer, Waltham, MA, USA) following anesthetization. To monitor tumor growth, the mice in the GVL treatment experiments were subjected to bioluminescence imaging every week. The *in vivo* imaging data were analyzed using Living Image software (Perkin Elmer).

## Flow cytometry

The animals were anesthetized with pentobarbital sodium (30 mg/kg intraperitoneally, i.p., Kyoritsu Seiyaku, Tokyo, Japan). The spleen and femur bone marrow were excised from the mice and minced into small pieces under deep anesthesia. The cell debris was removed using a 40-µl cell strainer. The number of cells was then counted. These cells were resuspended in FACS buffer and stained as previously described using the following antibodies: allophycocyanin (APC)-CY7-CD45 (HI30) (Bio Legend, San Diego, CA), Peridinin-chlorophyll-protein complex (PerCP)-CY 5.5-CD3 (UCHT1) (BD Biosciences, Franklin Lakes, NJ), APC-CD4

(RPA-T4) (eBioscience, San Diego, CA), fluorescein isothiocyanate (FITC)-CD8 (HIT8a) (BD Biosciences), Brilliant Violet (BV) 421-CD25 (M-A251) (BD Biosciences), phycoerythrin (PE)-FOXP3 (PCH101) (BD Biosciences), BV711-CD19 (SJ25C1) (BD Biosciences), and PE-Cy7-anti-mouse monoclonal CD45 (30-F11) (BD Biosciences) [4]. Dead cells were identified by BioFixable Viability Dye eFlour 660 (eBioscience). Fc-block was used to prevent the non-specific binding of antibodies to Fc receptors. Samples were acquired using BD LSRFortessa (BD Biosciences) and analyzed with FlowJo software (FlowJo LLC, Ashland, OR).

## Histopathology and immunohistochemistry

The left lung, liver, spleen, kidney, skin, and intestinal specimens were fixed in 4% paraformaldehyde with 0.05 M phosphate buffer (pH 7.4) overnight at 4˚C after perfusion fixation and then embedded in paraffin. Sections (4-µm-thick) were prepared. After the slides were deparaffinized, they were subjected to staining with H&E, terminal deoxynucleotidyl transferase dUTP nick end labeling (TUNEL), or immunohistochemistry. TUNEL staining was performed using an *in situ* Apoptosis Detection Kit (Takara Biochemicals, Shiga, Japan) in accordance with the manufacturer's instructions. For the immunohistochemistry analysis, the antigens were retrieved using the following agents: ×200 Immunosaver (Nissin EM, Tokyo, Japan) and/or citrate buffer (pH 6.0) at 95˚C for 60 minutes. These sections were incubated with primary antibodies overnight following the blocking step for 30 minutes using 2% normal goat serum containing PBS at room temperature. The following primary antibodies were used: anti-human CD3 epsilon (C3e/1308) (Novus Biologicals Littleton, CO), CD45 (M0701), (Dako Glostrup, Denmark) Ki-67 (20Raj1) (eBioscience), and HRP conjugated anti-mouse B220 (RA3-6B2) (Santa Cruz Biotechnology, Santa Cruz, CA). To visualize the target antigen, the ABC (Vector Laboratories, CA, USA)–DAB (Nakarai Chemical, LTD, Kyoto, Japan) reaction for light field scanning and a TSA System (Green; Fluorescein and red; Cyanine 3, Perkin Elmer) for fluorescent double-immunohistochemistry were utilized in accordance with the manufacturer's instructions. The sections were scanned with an optical microscope (BX-63; Olympus Japan Co., Tokyo, Japan) or a confocal laser microscope (FV-1000; Olympus). Positive cells were counted and are presented as the number per square µm.

## ELISA

An ELISA kit for human IFN-γ was purchased from eBioscience and used according to the manufacturer's instructions.

## Statistical analysis

Differences between the two populations were evaluated with Fisher's exact test. Overall survival was evaluated with a Kaplan-Meier analysis. A p value of $<0.05$ was considered statistically significant. All statistical tests were performed with EZR (Saitama Medical Center, Jichi Medical University, Saitama, Japan) [7].

## Results

First, a xenogeneic GVHD mouse model was used to identify the characteristics of the three immunosuppressants (ATG, alemtuzumab, and PTCY) at doses equivalent to those administered to humans. The mice were divided into five groups as follows: irradiation-alone (without GVHD or immunosuppressants), GVHD without treatment (control), and GVHD treated with ATG, alemtuzumab, or PTCY at the same dosages as the clinically used administration methods for human patients shown in Fig 1A. We used the general clinical used dosages for

GVHD prophylaxis in human transplantation. We used the general clinical used dosages for GVHD prophylaxis in human transplantation. All the immunosuppressants were administered in two equally divided doses. Alemtuzumab at 0.5 mg/kg (0.25 mg/kg for each day) [8], and ATG at 10 mg/kg (5 mg/kg for each day) [9, 10] were administered on 3 and 4 days prior to transplantation, and cyclophosphamide at 100mg/kg (50 mg/kg for each day) were administered 3 and 4 days after transplantation. On the day of transplantation, the mice received the hPBMCs at a concentration of $5 \times 10^6$ cells following irradiation.

## 10 mg/kg of ATG completely ameliorated acute xenogeneic GVHD whereas 0.5 mg/kg of alemtuzumab and 100 mg/kg of PTCY had a limited GVHD prophylactic effect

The survival of the mice is shown in Fig 1B. The body weight in all mice decreased during the first week after transplantation due to irradiation. All mice in the control GVHD group died

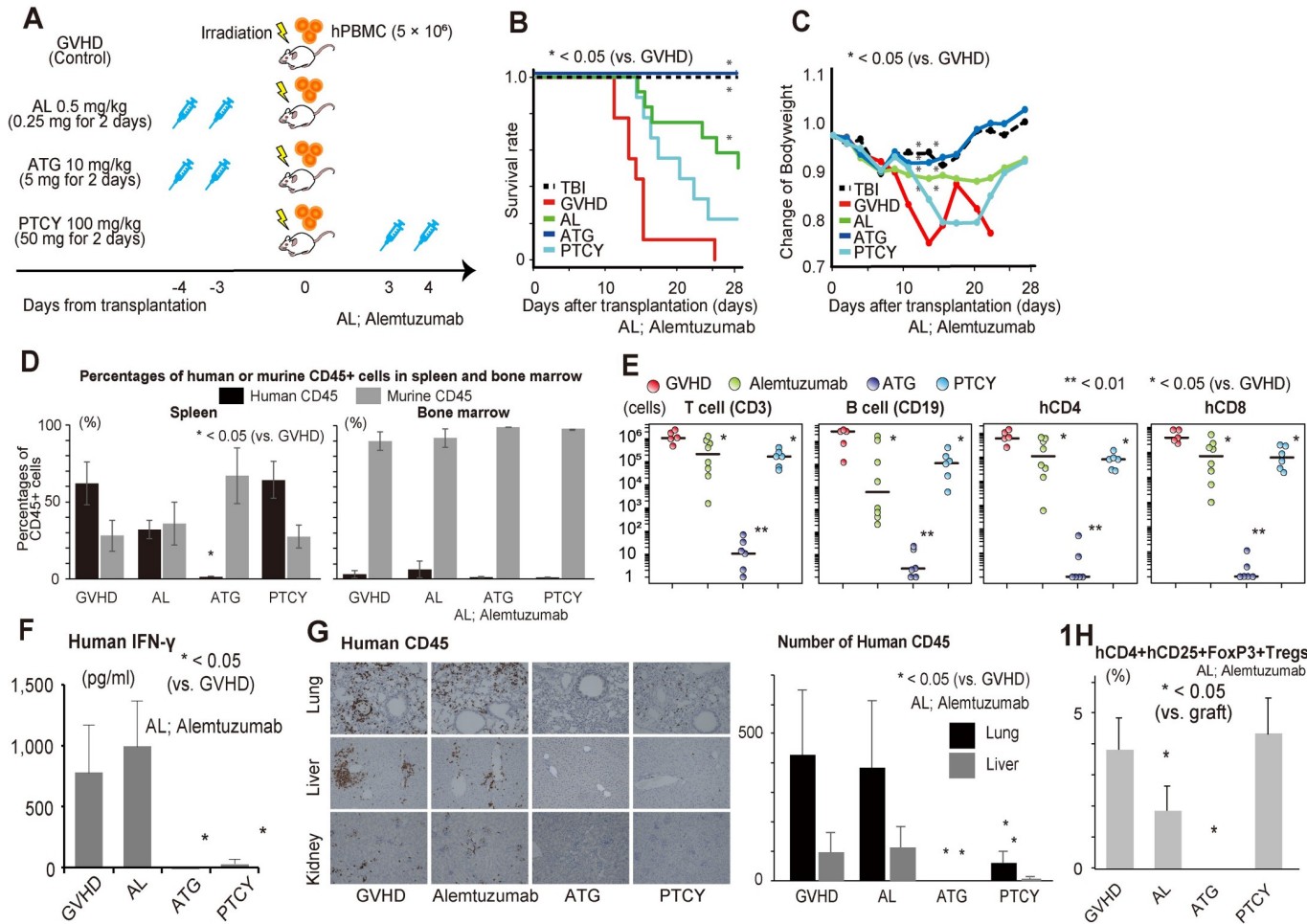

**Fig 1. Severity of GVHD and characteristics of GVHD inducing human cells among the ATG, alemtuzumab, and PTCY treatment groups.** Irradiated mice were administered hPBMCs with or without immunosuppressive drugs (A). We compared the survival curves (B) and body weight (C) among GVHD mouse models with/without immunosuppressive treatment (n = 9–12). The human and mouse hematopoietic cell composition in the spleen and bone marrow were measured in each model using flow cytometry (n = 5–8) (D). Absolute number of infiltrated human T/B cells in the spleens of mice were analyzed using flow cytometry (E). Serum levels of IFN-γ were measured in each model (F). Infiltration of human CD45-positive cells was mainly observed in the lung, liver, and kidney in this xenogeneic GVHD treatment mouse model (n = 6) (G). The proportion of human Tregs in infiltrated human CD4 T cells was analyzed using flowcytometry (H).

within 25 days after transplantation whereas none in the TBI-alone group died during this time period; neither group displayed signs of GVHD (Fig 1C). The ATG and alemtuzumab groups showed prolonged survival compared to the control GVHD group; this improvement was not significant in the PTCY group. Similar results were observed regarding the change in BW; the percentage weight loss was the highest in the PTCY group, followed by the alemtuzumab group. The ATG group exhibited less severe GVHD than the control GVHD group and had a similar body weight as the irradiation-alone group. These results suggested that 10 mg/kg of ATG ameliorated GVHD completely whereas 0.5 mg/kg of alemtuzumab and 100 mg/kg of PTCY had limited effects for preventing GVHD.

Next, we compared the phenotypes of infiltrated human cells in mouse organs using the same models. hPBMCs were mainly observed in the spleen but not in the bone marrow in mice according to the FACS analysis (Fig 1D). The 10 mg/kg of ATG group showed a markedly reduced infiltration of hPBMCs into the spleen and BM (bone marrow), which supported their prolonged survival without signs of GVHD, in contrast to the control group. Conversely, the alemtuzumab group had a large number of infiltrated hPBMCs despite their prolonged survival. Fig 1E shows the number and phenotypes of infiltrated hPBMCs in the spleen for each treatment group. The numbers of T and B cells were markedly reduced in the ATG group but not in the alemtuzumab and PTCY groups compared to the GVHD group. The serum IFN-γ level was also significantly decreased in the ATG and PTCY groups compared to the GVHD group (Fig 1F). In contrast to the ATG and PTCY treatments, alemtuzumab treatment did not reduce IFN-γ production despite increasing survival.

To further confirm these findings, we compared the histopathologies of other GVHD target organs, such as the lungs, liver, kidney, and intestines. The infiltration of human CD45-positive cells was primarily observed in the lungs, liver, and kidney in this xenogeneic GVHD treatment mouse model (Fig 1G). Large numbers of human CD3 cells infiltrating GVHD target organs were observed in the alemtuzumab group. To elucidate the cause of this observation, we assessed the apoptosis and proliferation of hCD3 by TUNEL staining and Ki-67 staining, respectively (S1 Fig). However, we did not find any significant differences in the proportions of TUNEL-positive cells and Ki-67-positive cells between the alemtuzumab and GVHD groups. The infiltrated cells seemed to be in the growth phase, and few showed apoptosis.

Using splenocytes, we also analyzed the percentages of human regulatory T cells (Tregs), which are known to suppress human GVHD. The proportions of Foxp3-expressing CD25 + CD4 + regulatory T cells in the alemtuzumab and ATG groups were significantly lower than those in the GVHD group (Fig 1H). In contrast to previous reports on human transplantation, PTCY did not increase the proportion of Tregs in this model. As shown above, alemtuzumab treatment resulted in prolonged survival, which is inconsistent with donor T cell proliferation in GVHD target organs without an increase in the proportion of Tregs.

However, these findings might have strongly depended on the doses of the drugs rather than their properties, and the doses used clinically for humans might not be appropriate in this mouse model. Although the immunosuppressive efficacy may be increased with higher doses, excessive immunosuppression may impair the GVL effect. Therefore, we determined the optimal dosage of the immunosuppressants for the following xenogeneic GVL/GVHD model.

## Establishment of a GVL treatment model using immunosuppressants

Next, we established a leukemia model with luciferase-transfected A20 cells (Luc-A20) and an IVIS imaging system. Tumor growth was detected using bioluminescence imaging (S2B Fig). A histopathological analysis identified A20 cell invasion in the bone marrow, liver, and spleen,

but not in the intestine or kidney (S2C Fig). Without the co-infusion of hPBMCs, the leukemia mice died approximately 30 days after transplantation with a marked increase in luciferase activity. Conversely, the leukemia mice with co-transplanted hPBMCs (GVL model) exhibited the complete eradiation of A20 leukemic cells but died with GVHD (S2A and S2B Fig). Although the GVL model showed many infiltrating hCD3 cells in the mouse tissue, there was no histopathological evidence of A20 cell invasion into the mouse organs in the GVL group (S2D Fig). Using this GVL model, we titrated the dosages of ATG, alemtuzumab, and PTCY in the following experiments.

## Establishment of an ATG model

An ATG dose of 10 mg/kg completely depleted the hPBMCs in the GVHD experiment; therefore, we reduced the ATG dosages to 5, 2.5, and 1.25 mg/kg in the GVL model. The ATG group showed better survival than the GVHD and GVL models (Fig 2A). Bioluminescence imaging showed that none of the mice in the GVHD and GVL groups exceeded $10^6$ photons per second, whereas the mice in the leukemia group died with more than $10^7$ photons per

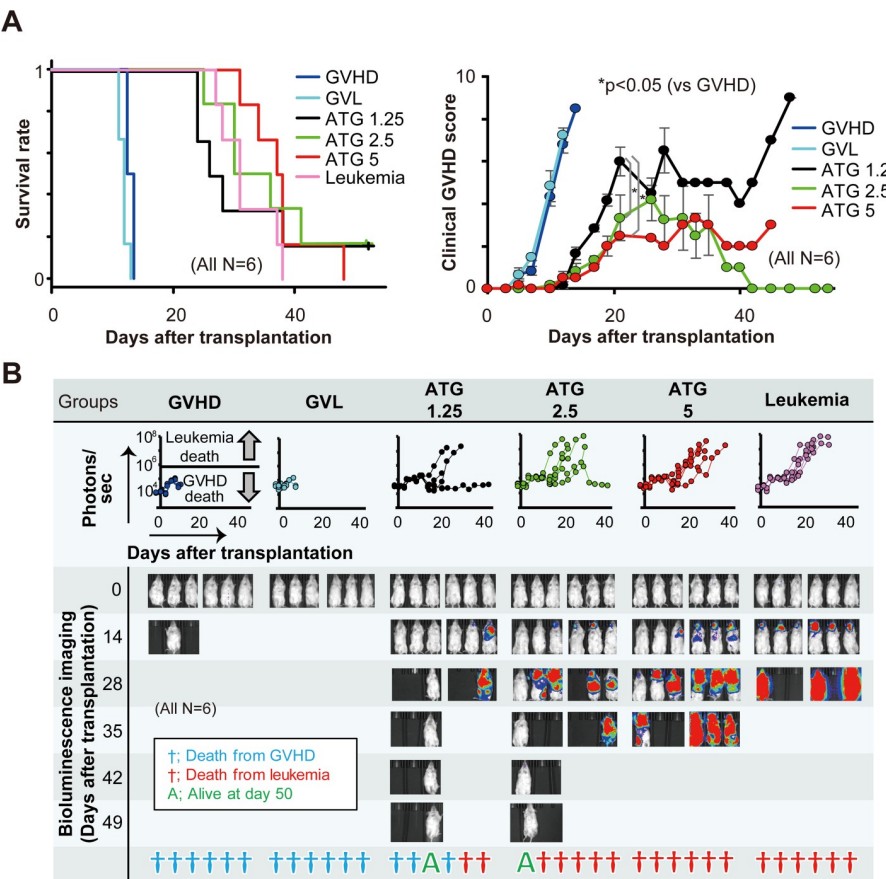

**Fig 2. Comparison of survival, clinical GVHD score, and tumor burden among GVHD, GVL with/without ATG treatment (1.25, 2.5, and 5 mg/kg), and leukemia mouse models.** We allocated the mice into six groups: GVHD, GVL, leukemia, and ATG at doses of 1.25, 2.5, and 5 mg/kg (n = 6 each). The mice were observed every day following transplantation for survival estimates and every other day to calculate their clinical GVHD scores (A). Bioluminescence imaging with photons for each group and the status of the mice were observed for 50 days following transplantation (B). The line graphs in the middle row show the whole-body bioluminescence photons. Blue, red, or green symbols represent the status of the mice at day 50 for GVHD death, leukemia death, or survival, respectively.

second (Fig 2B). Two of the six mice in the 1.25 mg/kg of ATG group, five of six mice in the 2.5 mg/kg of ATG group, and all the mice in the 5 mg/kg of ATG group developed leukemia. The remaining mice died from GVHD. The histopathological analysis showed that hPBMCs were decreased in a dose-dependent manner, and nearly all hPBMCs were depleted in the 5 mg/kg of ATG group (S3 Fig). Residual hPBMCs were observed in the tissues of the 2.5 mg/kg of ATG-treated mice, but 83% of them developed leukemia. Therefore, we considered the optimal ATG dosage for reducing GVHD without losing a GVL effect to range between 1.25 and 2.5 mg/kg. Thus, we subsequently allocated the mice into three further groups: ATG treatment at dosages of 1.5, 1.75, and 2.0 mg/kg (S4 Fig). However, all the mice died from GVHD or leukemia. These results suggest that the therapeutic window for ATG might be very narrow, if present at all.

## Establishment of an alemtuzumab model

We compared the effects of 1.0, 0.5, and 0.25 mg/kg of alemtuzumab in the GVL model. As shown in Fig 3A, survival in the 0.25 mg/kg of alemtuzumab group was not significantly different than that in the groups with GVHD or GVL without any treatment. None of the mice in the 0.5 mg/kg of alemtuzumab group and two-thirds of those in the 1 mg/kg group developed

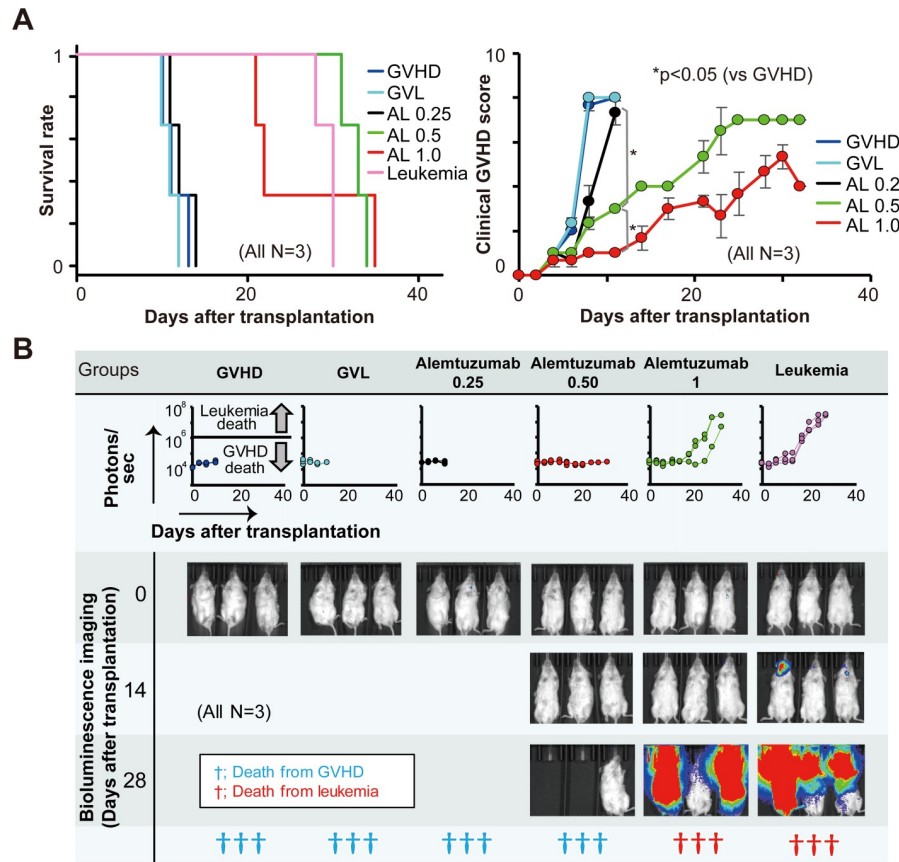

**Fig 3. Comparison of survival, clinical GVHD score, and tumor burden among GVHD, GVL with/without alemtuzumab treatment (0.25, 0.5, and 1.0 mg/kg), and leukemia mouse models.** The mice were allocated to one of the following six groups: GVHD, GVL, leukemia, and alemtuzumab treatment at doses of 0.25, 0.5, and 1.0 mg/kg (n = 3 each). The mice were observed every day following transplantation for survival estimates and every other day to calculate their clinical GVHD scores (A). Bioluminescence imaging with photons (line graphs in the middle row) for each group and the status of the mice were observed for 50 days following transplantation (B).

leukemia, and the remaining mice died from GVHD (Fig 3B). No leukemia cells were observed in the 0.25 and 0.5 mg/kg of alemtuzumab groups, and hPBMCs remained even in the tissues of the 1.0 mg/kg of alemtuzumab group (S3 Fig). Therefore, the appropriate dosage of alemtuzumab for this mouse model seemed to range between 0.5 and 1 mg/kg. We next allocated the mice into three groups: alemtuzumab treatment at doses of 0.6, 0.75, and 0.9 mg/kg (S5 Fig). The survival and severity of GVHD did not differ significantly among the three groups, with similar proportions of leukemic death and GVHD death, respectively. Therefore, the optimal dosage of alemtuzumab for this mouse model might be approximately 0.6 to 0.9 mg/kg, but, similar to ATG, the therapeutic window of alemtuzumab seemed to be narrow.

## Establishment of a PTCY model

We compared 100, 200, and 400 mg/kg of PTCY in this GVL model. As shown in Fig 4A and 4B, although none of the treatment groups developed leukemia, 83% of the 100 mg/kg of PTCY group and 67% of the 200 mg/kg of PTCY group died from GVHD within 50 days after transplantation. We decided that the optimal dosage of PTCY was 400 mg/kg, at which the model mice never developed leukemia or GVHD.

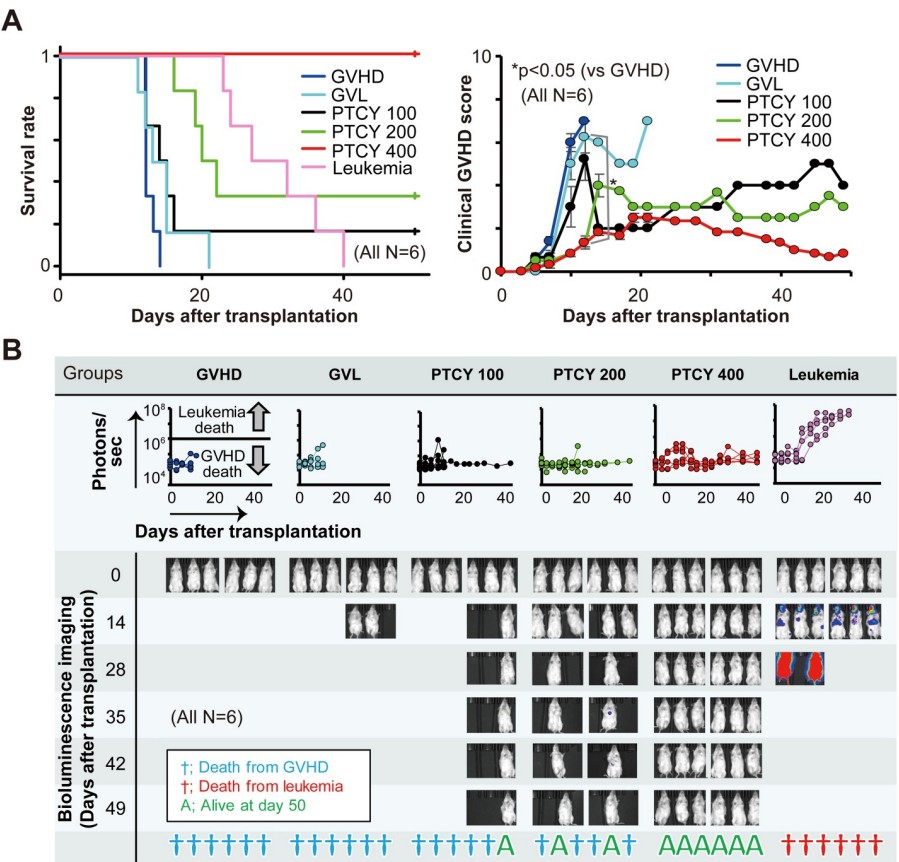

**Fig 4. Comparison of survival, clinical GVHD score, and tumor burden among GVHD, GVL with/without PTCY treatment (100, 200, and 400 mg/kg) and leukemia mouse models.** The mice were allocated to one of the following six groups: GVHD, GVL, leukemia, and PTCY at doses of 100, 200, and 400 mg/kg (n = 6 each). The mice were observed daily for survival following transplantation and every other day to calculate their clinical GVHD scores (A). Bioluminescence imaging with photons (line graphs in the middle row) for each group and the status of the mice were observed for 50 days following transplantation (B).

## Discussion

HLA-mismatched haploidentical transplantation is an alternative treatment for patients who do not have an HLA-matched donor. Immunosuppressive drugs such as ATG, alemtuzumab, and PTCY have been used to prevent GVHD in HLA-mismatched haploidentical HSCT. However, no clinical trials have directly compared these three drugs. In this study, we used xenogeneic GVHD mouse models and compared three types of immunosuppressive drugs to investigate whether these well-known immunosuppressants could ameliorate GVHD without impairing the GVL effect. Since ATG and alemtuzumab are anti-human lymphocyte antibodies, we used xenogeneic GVHD mouse models.

First, we administered these drugs to GVHD mouse models at dosages used clinically. ATG at a dosage of 10 mg/kg completely depleted hPBMCs and prolonged survival without evidence of GVHD. Moreover, in a GVL experiment, even an ATG dose of 5 mg/kg depleted human cells completely. Conversely, although 33% of the 0.5 mg/kg of alemtuzumab group and 67% of the 100 mg/kg of PTCY group died due to GVHD in the GVHD experiment, none of the mice in these groups developed leukemia in the GVL experiment. Based on these results, the best immunosuppressant for the prophylaxis of GVHD was 5 mg/kg of ATG, whereas 0.5 mg/kg of alemtuzumab or 100 mg/kg of PTCY were the best for maintaining a GVL effect. However, these results largely depended on the drug dosages. Therefore, we next determined the appropriate dosages to suppress GVHD while maintaining a GVL effect. By using the A20 leukemia cell line with the co-infusion of hPBMCs, we found that the optimal therapeutic windows appeared to be 0.6–0.9 mg/kg for ATG, 1.5–2.0 mg/kg for alemtuzumab, and 400 mg/kg for PTCY under our experimental conditions.

There have been no head to head randomized trial comparing of alemtuzumab, ATG, and PTCY until today. In 2019, Battipaglia et al. reported large-scale retrospective analysis of outcome comparison between ATG and PTCY (50 mg/kg in general) for HLA-mismatched unrelated donor transplantation [11]. According to them, PTCY was significantly decreased the incidence of grade III-IV acute GVHD at 100 days compared to ATG at average dosage of 6mg/kg (9% vs 19%, $P < 0.04$). However, it is difficult to determine the optimal dosage of ATG. Another report also showed that PTCY had lower acute GVHD incidence than ATG 6 mg or 2.5 mg/kg in unrelated donor transplantation [12]. Therefore, PTCY at 50 mg/kg has built a strong position as a safest and most economical prophylaxis method against GVHD. However, an optimal dosage of ATG remains still controversial. Some said that personalized ATG dosage for GVHD prophylaxis may improve outcomes after transplantation [13, 14]. In recent years, low dose therapies of alemtuzumab (0.5 mg/kg) [8] and ATG (2–6 mg/kg) have been developing [15–17]. The dosage of ATG varied depending on the studies, and individual dosing was reported to improve reconstitution of CD4 T cell and optimal dosage may differ among races. As we demonstrated, the immunosuppressive effect of ATG and alemtuzumab strongly depended on their dosages, so that the dose titration studies especially in low dose range in clinical setting are needed.

Tregs are well known to suppress the proliferation and cytokine production of T cells and the alloimmune response. They also suppressed GVHD in a mouse model without reducing the GVL effect [18]. It remains controversial whether immunosuppressive drugs for GVHD prophylaxis can induce or increase the number of Tregs. Some reports have shown that Tregs were induced by the administration of ATG but not alemtuzumab [19–21]. Another report showed that campath-1H and another anti-CD52 monoclonal antibody with anti-CD3 stimulation induced Tregs in vitro [22]. However, Tregs were not expanded by immunosuppressive drugs in our in vivo study. These discrepancies among previous reports and our data suggest that allo- and xenogeneic immune stimulation may not be sufficient to expand Tregs.

The clinically used doses of human antibody drugs were excessive in this mouse model. Antibody drugs are known to be mainly eliminated by receptor-mediated endocytosis, on which depends on the expression of the target and affinity of the antibody [23]. Moreover, the concentration of alemtuzumab is known to be affected by the target cell count and decreases rapidly in patients with CD52-expressing tumors [24, 25]. Therefore, particularly in the ATG and alemtuzumab groups, these mouse models required greater amounts of immunosuppressants than the dosages used clinically. However, the optimal dosages of ATG and alemtuzumab were lower than the doses used clinically for humans. This suggests that a considerable amount of antibody drugs are associated with recipient hematopoietic cells and tissues in human HSCT.

The irradiated NOG mice received $5 \times 10^6$ hPBMCs per mouse, which is equivalent to $10^9$ cells per kilogram and almost the same amount of CD3-positive cells in collected PBMCs for allo-transplantation. Therefore, this model might be more appropriate for evaluating GVL than GVHD because the lymphocytes that could strongly attack tumor cells are thought to be injected T cells rather than differentiated donor cells after engraftment. The optimal dose of immunosuppressants is the dose at which GVHD is prevented without reducing GVL. However, the therapeutic windows for ATG and alemtuzumab appeared to be very narrow in this study. All the mice in 400 mg/kg of PTCY group showed a long period of survival without leukemia or GVHD. However, cyclophosphamide has been shown to kill A20 cells [26], and therefore, this result does not indicate that 400 mg/kg of PTCY prevents GVHD while maintaining a GVL effect. Confirmatory experiments using tumor cells that are not sensitive to cyclophosphamide must be performed.

This study has several limitations. First, we used a xenogeneic model instead of an allogeneic model. We have previously shown that the profile of GVHD in a xenogeneic GVHD model is somewhat different from that in human GVHD [4]. Second, the drugs might be metabolized differently in humans and mice. For example, although we were unable to monitor the blood concentration of alemtuzumab in this model, previous reports showed a long terminal half-life of alemtuzumab [27, 28]. The longer half-life of alemtuzumab might cause the long-term suppression of T cells, and this may promote the suppression of GVHD while suppressing the GVLD effect.

## Conclusions

In conclusion, we compared the effects of ATG, alemtuzumab, and PTCY on GVHD and GVL in xenogeneic GVHD models. The optimal therapeutic windows under the current conditions might be very narrow, and further studies are warranted to maximize the benefits of each immunosuppressant.

## Supporting information

**S1 Fig. Analysis of human cell proliferation and apoptosis using immunohistochemistry staining.** Sections of lungs and liver in the mice were stained with human CD3, Ki-67, and TUNEL and detected with DAB. Methyl green was used for a nuclear counterstain. (TIF)

**S2 Fig. A20 leukemia and GVL mouse models.** $2 \times 10^3$ luciferase-transfected (Luc) A20 cells were transplanted into the irradiated mice to create the leukemia model, and $2 \times 10^3$ Luc-A20 cells with $5 \times 10^6$ hPBMCs were co-transplanted to create the GVL model. Each group consisted of three mice. They were observed daily to assess survival (A), and tumor growth was detected using bioluminescence imaging (B). The histopathological analysis also showed A20 tumor

growth. Sections of the bone marrow, liver, and spleen of the leukemia mouse model were stained with hematoxylin and eosin (C). Immunohistochemistry staining with anti-B220 (mouse B cell) and anti-human CD3 detected A20 tumor and hPBMC invasion, respectively (bone marrow, liver, and spleen) (D).
(TIF)

**S3 Fig. Fluorescent immunohistochemistry of human CD3 and mouse B220 in sections of the organs of mice.** Sections of the lungs and liver in mice were stained with B220 (Fluorescein, green) and human CD3 (Cyanine 3, red) and detected by fluorescent immunohistochemistry with tyramide signal amplification.
(TIF)

**S4 Fig. Comparison of survival and clinical GVHD scores among GVHD, GVL with/without ATG treatment (1.5, 1.75, and 2.0 mg/kg) and leukemia mouse models.** We compared ATG treatment at doses ranging from 1.25 to 2.5 mg/kg (1.5, 1.75, and 2.0 mg/kg). Each group consisted of six mice, and the mice were allocated to one of following six groups: GVHD, GVL with/without ATG treatment (1.5, 1.75, and 2.0 mg/kg), and leukemia. The mice were observed every day following transplantation for survival estimates and every other day to calculate their clinical GVHD scores (A). Bioluminescence imaging with photons (line graphs in the middle row) for each group and status of mice were observed for 50 days following transplantation (B).
(TIF)

**S5 Fig. Comparison of survival and clinical GVHD scores among GVHD, GVL with/without alemtuzumab treatment (0.6, 0.75, and 0.9 mg/kg), and leukemia mouse models.** We compared alemtuzumab treatment at doses ranging from 0.5 to 1.0 mg/kg (0.6, 0.75, and 0.9 mg/kg). The mice were allocated to one of the following six groups: GVHD, GVL, alemtuzumab at 0.6, 0.75, and 0.9 mg/kg, and leukemia. Each group consisted of six mice, and they were observed every day following transplantation for survival estimates and every other day to calculate their clinical GVHD scores (A). Bioluminescence imaging with photons (line graphs in the middle row) for each group and status of mice were observed for 50 days following transplantation (B).
(TIF)

## Acknowledgments

The authors would like to thank Dr. Morio Azuma and Dr. Hiroko Hayakawa for their helpful technical advice.

## Author Contributions

**Conceptualization:** Yoshinobu Kanda.

**Data curation:** Kiyomi Mashima, Nobuhiko Ohno, Yoshinobu Kanda.

**Formal analysis:** Kiyomi Mashima.

**Investigation:** Kiyomi Mashima, Iekuni Oh, Junko Izawa, Norihito Takayama, Hirofumi Nakano, Yasufumi Kawasaki, Nobuhiko Ohno, Yoshinobu Kanda.

**Methodology:** Kiyomi Mashima, Iekuni Oh, Ken Fujiwara, Junko Izawa, Norihito Takayama, Hirofumi Nakano, Yasufumi Kawasaki, Nobuhiko Ohno, Yoshinobu Kanda.

**Project administration:** Yoshinobu Kanda.

**Resources:** Yoshinobu Kanda.

**Supervision:** Iekuni Oh, Ken Fujiwara, Daisuke Minakata, Kaoru Morita, Masahiro Ashizawa, Chihiro Yamamoto, Kaoru Hatano, Kazuya Sato, Ken Ohmine, Shin-Ichiro Fujiwara, Nobuhiko Ohno, Yoshinobu Kanda.

**Writing – original draft:** Kiyomi Mashima, Yoshinobu Kanda.

**Writing – review & editing:** Iekuni Oh, Ken Fujiwara, Daisuke Minakata, Ryoko Yamasaki, Kaoru Morita, Masahiro Ashizawa, Chihiro Yamamoto, Kaoru Hatano, Kazuya Sato, Ken Ohmine, Shin-Ichiro Fujiwara, Nobuhiko Ohno, Yoshinobu Kanda.

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
