## [Decision Letter · Decision Letter 0]

6 Oct 2020

PONE-D-20-18979

Comparison of alemtuzumab, anti-thymocyte globulin, and post-transplant cyclophosphamide for graft-versus-host disease and graft-versus-leukemia in murine models

PLOS ONE

Dear Dr. Kanda,

Thank you for submitting your manuscript to PLOS ONE. After careful consideration, we feel that it has merit but does not fully meet PLOS ONE’s publication criteria as it currently stands. Therefore, we invite you to submit a revised version of the manuscript that addresses the points raised during the review process.

We have now received reports from two referees of your manuscript, as agree with reviewers comments raised a few concerns about this study. After careful consideration, we invite you to submit a revised version of the manuscript.  

We look forward to receiving your revised manuscript.

Kind regards,

Senthilnathan Palaniyandi, Ph.D

Academic Editor

PLOS ONE

Journal Requirements:

2. Please address the following:

- To comply with PLOS ONE submissions requirements, in your Methods section, please provide additional information on the animal research and ensure you have included details on (1) animal upkeep, (2) methods of anesthesia and/or analgesia, and (3) efforts to alleviate suffering.

- In your Methods section, please provide additional details regarding the cell lines used in your study and ensure you have described the source. For more information regarding PLOS' policy on materials sharing and reporting, see https://journals.plos.org/plosone/s/materials-and-software-sharing#loc-sharing-materials, and for more information on PLOS ONE's guidelines for research using cell lines, see https://journals.plos.org/plosone/s/submission-guidelines#loc-cell-lines.

"This work was supported in part by JKA through its promotion

funds from KEIRIN RACE, grants from the Ministry of Health, Welfare, and Labor of Japan,

and Grants-in-Aid for Scientific Research from the Ministry of Education, Science, Sports,

and Technology of Japan."

Reviewers' comments:

Reviewer's Responses to Questions

**Comments to the Author**

1. Is the manuscript technically sound, and do the data support the conclusions?

Reviewer #1: No

Reviewer #2: Yes

2. Has the statistical analysis been performed appropriately and rigorously? 

Reviewer #1: Yes

Reviewer #2: Yes

3. Have the authors made all data underlying the findings in their manuscript fully available?

Reviewer #1: No

Reviewer #2: Yes

4. Is the manuscript presented in an intelligible fashion and written in standard English?

Reviewer #1: Yes

Reviewer #2: Yes

5. Review Comments to the Author

Reviewer #1: This paper by Mashima and colleagues presents a mouse model to compare 3 clinically utilized drugs for the treatment of GVHD in patients undergoing allogeneic stem cell transplantation. While research into the differing effects of commonly used treatments for GVHD are required to refine their usage, this paper is disappointing in it's lack of detailed comparison of the 3 drugs investigated and draws no conclusions about the clinical application of these therapies. Rather, this paper reads as model development. The experimental design is unclear at times and it is unclear if the experiments reported are a single experiment or were repeated multiple times for scientific rigor.

Line 54 - please correct 'retrospective clinical trials' - clinical trials by their nature are prospective. The correct term is retrospective cohort studies or retrospective analysis.

Line 70 - please correct to take into account that cyclophosphamide is not a monoclonal antibody

Please provide complete human ethics approval details including that you have complied with the declaration of Helsinki.

Please provide details for how GVHD clinical scores were determined and any thresholds for humane killing when mice exhibited signs of distress (e.g. loss of body weight, high clinical score)

Please specify the supplier for each reagent rather than having a list of reagents and saying that they could have come from 3 suppliers. Please include flurophore for each specific antibody.

Line 170-173 - the way the dosing is written is confusing, please revise. Also provide details as to how these doses were chosen

Figure 1 - please include clinical GVHD scores

Figure 1D - this would be better presented as %CD45+ cells to account for differences in cellularity between mice

Line 222 - please revise - starting the sentence with 'Afterwards' suggests that the analysis was performed a long time after sacrifice but I assume it was done on fresh samples? Also please state what tissue was used for this analysis. This Treg data is possibly the most interesting part of the paper but it unfortunately is not repeated in subsequent experiments and therefore is of limited value.

Please explain the GVHD control in the experiments presented in figures 2-4. It is unclear how this mouse control was generated as if it wasn't injected with A20 cells (which would make it a GVL control) it won't have disease that can be detected in the IVIS.

Figures 2-4 show dose finding experiments for the 3 drugs in this study. Having completed this, it would have been good to see another head to head comparison as done in figure 1 to compare and contrast the effects of these 3 drugs, including histology, GVHD clinical scores and immune profiling in organs. Even more interesting would be examining if these drugs could be combined to improve GVL while reducing GVHD.

The discussion and conclusion needs to be revised to put the work in the context of the field including referencing other published works in the field.

While authors state data is available without restrictions in the information entered into the submission system. There is no data availability statement in the manuscript.

Reviewer #2: Mouse models for human leukemia and for human graft versus host disease are difficult. The authors show that within their model human ATG at doses of 1.25 - 2.5 mg/kg works best to ameliorate graft-versus host disease. Interestingly, post transplant cyclophosphamide which has a different mechanism of action was less effective. Interestingly, total body irradiation (used as control) was 100% effective in eliminating GVHD. There are many experiments which could have expanded this research (e.g. measuring other chemokines and cytokines, using other cell lines) but this article gives an idea how commonly used immunosuppressants work in a mouse model. The data appear well documented.

6. PLOS authors have the option to publish the peer review history of their article (what does this mean?). If published, this will include your full peer review and any attached files.

Reviewer #1: No

Reviewer #2: No

---

## [Author Response · Author response to Decision Letter 0]

20 Nov 2020

We are grateful to you and the reviewers for thoughtful comment that have helped us to improve this manuscript. Followings are our answers for the comments of the editor and reviewers.

According to the editor’s comments, we revised our manuscript as follows. 

Lines 85-90.

We performed our animal experiments as described before [6]. In brief, the animals were maintained under a 12-hr light/dark cycle and given conventional food and water ad libitum in 23℃ room. The animals were anesthetized with pentobarbital sodium (30 mg/kg intraperitoneally, i.p., Kyoritsu Seiyaku, Tokyo, Japan). They were euthanized when body weight loss reached 15% within a few days or an overall body weight loss reached 20%.

We also add the following sentences to Lines 76-80.

Firefly luciferase-transfected A20 BALB/c strain mouse B leukemia and lymphoma cells were kindly gifted by Dr. K. Ohnuma in October 2017. (Juntendo University, Tokyo, Japan) [5].

We removed funding information from our manuscript. The reason is that Fortessa we use for FACS analysis is university-owned equipment and JKA supported financially at the time of purchase, but it is unnecessary to disclose in our manuscript. 

We add a reference to support the sentence with “data not shown”.

Reviewer #1: This paper by Mashima and colleagues presents a mouse model to compare 3 clinically utilized drugs for the treatment of GVHD in patients undergoing allogeneic stem cell transplantation. While research into the differing effects of commonly used treatments for GVHD are required to refine their usage, this paper is disappointing in it's lack of detailed comparison of the 3 drugs investigated and draws no conclusions about the clinical application of these therapies. Rather, this paper reads as model development. The experimental design is unclear at times and it is unclear if the experiments reported are a single experiment or were repeated multiple times for scientific rigor.

Author’s comments

Thank you very much for the careful reading of our manuscript. We highly appreciate the reviewers' thoughtful and detailed comments with suggestions. 

We understand our experiment needed detail analysis of 3 drugs for in vivo GVHD mouse model with their optimal dosage. However, we did not repeat the GVHD experiment using the determined dosage based on GVL experiment for the following reason. 

In the PTCY group, since CY directly inhibited leukemia cells in this model, the final result was that PTCY at 400 mg/kg completely suppressed GVHD with elimination of leukemia. Therefore, we didn’t conclude PTCY at 400 mg/kg as the optimum concentration for GVHD prophylaxis. Also, in alemtuzumab and ATG groups, the optimum concentration, which completely suppressed GVHD without reducing GVL, was not determined. Therefore, in the end, the first GVHD experiment was not repeated based on the GVL experiment. 

We performed this experiment multiple times, not many mice at a time, because the limitation of this mouse model depended on the number of human PBMC from the healthy donor. The Review Board of Jichi Medical University determined that the maximum amount of blood allowed to draw was 40 ml per each donor. From the points of both scientific accuracy and limitation of the number of human PBMC, we repeated the experiment more than twice except for GVL experiment with alemtuzumab. Each time, we used three mice per each group and repeated the experiment two to three times. The experiments in all the groups were performed at the same time with a single healthy donor. We used different donors in the experiments at different time. 

According to the reviewer, we add the following sentences to Lines 96-99.

Each time, we used three mice per group and repeated the experiment two to three times. The experiments in all the groups were performed at the same time with a single healthy donor. We used different donors in the experiments at different time.

Line 54 - please correct 'retrospective clinical trials' – clinical trials by their nature are prospective. The correct term is retrospective cohort studies or retrospective analysis.

Author’s comments

We appreciate this comment. According to the reviewer’s comment, we changed the sentence (Line 54) as follows.

Some retrospective analysis have compared the immunosuppressive effects of alemtuzumab, ATG, and PTCY for GVHD prophylaxis.

Line 70 - please correct to take into account that cyclophosphamide is not a monoclonal antibody 

Author’s comments

I agree with this comment. According to the reviewer’s comment, we revised the sentence (Lines 69 – 70) as follows;

NOG mice to evaluate the effects of immunosuppressants against human cells.

Please provide complete human ethics approval details including that you have complied with the declaration of Helsinki.

Author’s comments

We apologize that our manuscript lacks the detailed description for the declaration of Helsinki. We add the following sentence to Lines 76 – 78.

This study was approved by the Institutional Review Board of Jichi Medical University and was conducted according to the principles of the Declaration of Helsinki.

Please provide details for how GVHD clinical scores were determined and any thresholds for humane killing when mice exhibited signs of distress 

(e.g. loss of body weight, high clinical score)

Author’s comments

We apologize for the lack of the information of the GVHD score. The clinical GVHD scores were calculated based on body weight BW, activity, skin, fur ruffing, and posture. Each factor received 0 to 2 scores, and the total score was determined by sum them (maximum index was 10.) as we previously reported (Kawasaki et al, Biology of Blood and Marrow Transplantation, 2018). Mice was euthanized when body weight loss reached 15% within a few days or an overall body weight loss reached 20%. According to the reviewer’s comment, we add the following sentences and references to Lines 89 and 119.

Lines 89 - 90 

They were euthanized when body weight loss reached 15% within a few days or an overall body weight loss reached 20%.

Line 119-121 

The clinical GVHD scores were calculated based on body weight, activity, skin, fur ruffing, and posture. Each factor received 0 to 2 scores, and the total score was determined by sum them (maximum index was 10.) as we previously reported [4].

Please specify the supplier for each reagent rather than having a list of reagents and saying that they could have come from 3 suppliers.

Please include flurophore for each specific antibody.

Author’s comments

We appreciate the reviewer’s comment. We corrected the following sentences to Lines 136 – 142 and Lines 158 - 160.

Lines 136 – 142

 allophycocyanin (APC)-CY7-CD45 (HI30) (Bio Legend, San Diego, CA), Peridinin-chlorophyll-protein complex (PerCP)-CY 5.5-CD3 (UCHT1) (BD Biosciences, Franklin Lakes, NJ), APC-CD4 (RPA-T4) (eBioscience, San Diego, CA), fluorescein isothiocyanate (FITC)-CD8 (HIT8a) (BD Biosciences), Brilliant Violet (BV) 421-CD25 (M-A251) (BD Biosciences), phycoerythrin (PE)-FOXP3 (PCH101) (BD Biosciences), BV711-CD19 (SJ25C1) (BD Biosciences), and PE-Cy7-anti-mouse monoclonal CD45 (30-F11) (BD Biosciences) [4].

 Lines 158-160

 anti-human CD3 epsilon (C3e/1308) (Novus Biologicals Littleton, CO), CD45 (M0701), (Dako Glostrup, Denmark) Ki-67 (20Raj1) (eBioscience), and HRP conjugated anti-mouse B220 (RA3-6B2) (Santa Cruz Biotechnology, Santa Cruz, CA).

Line 170-173 - the way the dosing is written is confusing, please revise. Also provide details as to how these doses were chosen

Author’s comments

We appreciate the reviewer’s comment. we agree with the reviewer that our description for dosing were complicated. Therefore, we add a new figure of the schema of our experiment (Fig 1A).

Since transplantation is high-risk treatment, there has been no standard method to prevent GVHD and it’s difficult to perform randomized control trial of immunosuppressants in real patients. Therefore, we compared three major GVHD prophylaxis methods with general clinical used dosages and time setting for humans in mice. According to the reviewer’s comment, we revised and add the following description to Lines 180-190. 

Lines 180-190

The mice were divided into five groups as follows: irradiation-alone (without GVHD or immunosuppressants), GVHD without treatment (control), and GVHD treated with ATG, alemtuzumab, or PTCY at the same dosages as the clinically used administration methods for human patients shown in Fig. 1A. We used the general clinical used dosages for GVHD prophylaxis in human transplantation. We used the general clinical used dosages for GVHD prophylaxis in human transplantation. All the immunosuppressants were administered in two equally divided doses. Alemtuzumab at 0.5 mg/kg (0.25 mg/kg for each day) [8], and ATG at 10 mg/kg (5 mg/kg for each day) [9, 10] were administered on 3 and 4 days prior to transplantation, and cyclophosphamide at 100mg/kg (50 mg/kg for each day) were administered 3 and 4 days after transplantation. On the day of transplantation, the mice received the hPBMCs at a concentration of 5 × 106 cells following irradiation.

Figure 1 - please include clinical GVHD scores

Author’s comments

We appreciate this comment and we totally agree that we also should have checked the clinical GVHD scores. However, we didn’t check the clinical GVHD scores in this first GVHD experiment and only checked body weight. Since this first experiment showed bigger differences among the groups than we expected, we proceeded to the next flowcytometry experiment. 

Figure 1D - this would be better presented as %CD45+ cells to account for differences in cellularity between mice

Author’s comments

We thank the reviewer for this suggestion. However, the sizes of the spleen were different between the groups (We didn’t take photographs.) and the total cell counts of spleen were shown in this Table. 

Absolute cell Number of splenocytes Average

GVHD 3200000 8000000 4320000 14300000 14300000 8824000

AL 1320000 28600000 1940000 750000 1600000 4500000 1400000 1400000 2400000 4878888

ATG 180000 520000 760000 340000 680000 970000 575000

CY 3040000 460000 1280000 700000 1000000 1100000 1263333

AL: Alemtuzumab

Human cells and mouse splenocytes increased under GVHD inflammatory condition, and immunosuppressants were able to reduce the proliferation of lymphocytes. Therefore, we assume that the percentages of human cell didn’t reflect the exact effect of immunosuppressive drugs, and the absolute number of each cell fraction was suitable for this comparison. 

Line 222 - please revise - starting the sentence with 'Afterwards' suggests that the analysis was performed a long time after sacrifice but I assume it was done on fresh samples? Also please state what tissue was used for this analysis. This Treg data is possibly the most interesting part of the paper but it unfortunately is not repeated in subsequent

experiments and therefore is of limited value.

Author’s comments

We apologize for the confusion. The answer is yes and we performed Treg analysis using fresh spleen sample. We corrected the sentences in Lines 239-240 as follows.

Using splenocytes, we also analyzed the percentages of human regulatory T cells (Tregs), Which are known to suppress human GVHD. 

In addition, we agree with the reviewer’s comment and this is one of the most interesting part of this experiment. However, we did not repeat the GVHD experiment based on the determined dosage in GVL experiment, because it was difficult to definite the optimal dosage in GVL experiment as explained above.

Please explain the GVHD control in the experiments presented in figures 2-4. It is unclear how this mouse control was generated as if it wasn't injected with A20 cells (which would make it a GVL control) it won't have disease that can be detected in the IVIS.

Author’s comments

We appreciate this comment. As we described above, we repeated our experiments with different donors, and survival varies in GVHD model depend on the individual donors. We simultaneously performed survival/GVHD score observation and IVIS imaging experiment. Therefore, we used GVHD model, in which no immunosuppressants were used, as a control group.

Figures 2-4 show dose finding experiments for the 3 drugs in this study. Having completed this, it would have been good to see another head to head comparison as done in figure 1 to compare and contrast the effects of these 3 drugs, including histology, GVHD clinical scores and immune profiling in organs. Even more interesting would be examining if these drugs could be combined to improve GVL while reducing GVHD.

Author’s comments

We appreciate this comment. We understand our experiment needed detail analysis of 3 drugs for in vivo GVHD mouse model with their optimal dosage. However, we did not repeat the GVHD experiment using the determined dosage based on GVL because we weren’t able to determine the optimal dosage of these 3 drugs from GVL experiment. In the PTCY group, since CY directly inhibited leukemia cells in this model, the final result was that PTCY at 400 mg/kg completely suppressed GVHD with elimination of leukemia. Therefore, we didn’t conclude PTCY at 400 mg/kg as the optimum concentration for GVHD prophylaxis. Also, in alemtuzumab and ATG groups, the optimum concentration, which completely suppressed GVHD without reducing GVL, was not determined. Therefore, in the end, the first GVHD experiment was not repeated based on the GVL experiment. 

The discussion and conclusion needs to be revised to put the work in the context of the field including referencing other published works in the field.

Author’s comments

We appreciate this comment and we add the following sentences to paragraph 3 in discussion part.

There have been no head to head randomized trials that compared alemtuzumab, ATG, and PTCY until today. In 2019, Battipaglia et al. reported large-scale retrospective analysis of outcome comparison between ATG and PTCY (100 mg/kg in general) for HLA-mismatched unrelated donor transplantation [8]. According to them, PTCY was significantly decreased the incidence of grade III-IV acute GVHD at 100 days compared to ATG at an average dosage of 6mg/kg (9% vs 19%, P < 0.04). However, it is difficult to determine the optimal dosage of ATG. Another report also showed that PTCY had lower acute GVHD incidence than ATG at 6 mg or 2.5 mg/kg in unrelated donor transplantation [9]. Therefore, PTCY at 100 mg/kg has built a strong position as the safest and most economical prophylaxis method against GVHD. However, the optimal dosage of ATG remains controversial. Some suggested that personalized ATG dosage for GVHD prophylaxis may improve outcomes after transplantation [10, 11]. In recent years, low-dose alemtuzumab (0.5 mg/kg) [12] and ATG (2 – 6 mg/kg) have been investigated [13-15]. The dosage of ATG varied depending on the studies, and individual dosing was reported to improve reconstitution of CD4 T cell and the optimal dosage may differ among races. As we demonstrated, the immunosuppressive effect of ATG and alemtuzumab strongly depended on their dosages, so that the dose titration studies especially in low dose range in clinical settings are needed.

While authors state data is available without restrictions in the information entered into the submission system. There is no data availability statement in the manuscript.

Author’s comments

We apologize for the lack of description about data availability. We will upload all the data sets including FCS files from FACS, survival, body weight changes, GVHD scores and so on. According to the reviewer, we add the following statement to Lines 425-426.

All relevant data are within the paper and its Supporting information files (https://figshare.com/articles/dataset/Supplementary_files/13256750).

 

Reviewer #2: Mouse models for human leukemia and for human graft versus

host disease are difficult. The authors show that within their model human ATG at doses of 1.25 - 2.5 mg/kg works best to ameliorate graft-versus host disease. Interestingly, post transplant cyclophosphamide which has a different mechanism of action was less effective. Interestingly, total body irradiation (used as control) was 100% effective in eliminating GVHD. There are many experiments which could have expanded this research (e.g. measuring other chemokines and cytokines, using other cell lines) but this article gives an idea how commonly used immunosuppressants work in a mouse model. The data appear well documented.

Author’s comments

Thank you very much for the careful reading of our manuscript. We revised our manuscript according to the reviewer’s comments. Again, we highly appreciate your taking time to review our manuscript. Thank you.

---

## [Decision Letter · Decision Letter 1]

26 Dec 2020

Comparison of alemtuzumab, anti-thymocyte globulin, and post-transplant cyclophosphamide for graft-versus-host disease and graft-versus-leukemia in murine models

PONE-D-20-18979R1

Dear Dr. Kanda,

We’re pleased to inform you that your manuscript has been judged scientifically suitable for publication and will be formally accepted for publication once it meets all outstanding technical requirements.

Kind regards,

Senthilnathan Palaniyandi, Ph.D

Academic Editor

PLOS ONE

Additional Editor Comments (optional):

Reviewers' comments:

Reviewer's Responses to Questions

**Comments to the Author**

1. If the authors have adequately addressed your comments raised in a previous round of review and you feel that this manuscript is now acceptable for publication, you may indicate that here to bypass the “Comments to the Author” section, enter your conflict of interest statement in the “Confidential to Editor” section, and submit your "Accept" recommendation.

Reviewer #1: All comments have been addressed

Reviewer #2: All comments have been addressed

2. Is the manuscript technically sound, and do the data support the conclusions?

Reviewer #1: (No Response)

Reviewer #2: Yes

3. Has the statistical analysis been performed appropriately and rigorously? 

Reviewer #1: (No Response)

Reviewer #2: Yes

4. Have the authors made all data underlying the findings in their manuscript fully available?

Reviewer #1: (No Response)

Reviewer #2: Yes

5. Is the manuscript presented in an intelligible fashion and written in standard English?

Reviewer #1: (No Response)

Reviewer #2: Yes

6. Review Comments to the Author

Reviewer #1: (No Response)

Reviewer #2: Mouse models for human graft-versus host disease are tricky. This manuscript is a useful beginning and should be published.

7. PLOS authors have the option to publish the peer review history of their article (what does this mean?). If published, this will include your full peer review and any attached files.

Reviewer #1: No

Reviewer #2: No

---

## [Editor Report · Acceptance letter]

2 Jan 2021

PONE-D-20-18979R1 

Comparison of alemtuzumab, anti-thymocyte globulin, and post-transplant cyclophosphamide for graft-versus-host disease and graft-versus-leukemia in murine models 

Dear Dr. Kanda:

I'm pleased to inform you that your manuscript has been deemed suitable for publication in PLOS ONE. Congratulations! Your manuscript is now with our production department. 

Kind regards, 

on behalf of

Dr. Senthilnathan Palaniyandi 

Academic Editor

PLOS ONE